# Evaluating the impact of test-trace-isolate for COVID-19 management and alternative strategies

**Kun Zhang[1], Zhichu Xia[2], Shudong Huang[1], Gui-Quan Sun[3,4], Jiancheng Lv[1], Marco Ajelli[5]☯, Keisuke Ejima[6]☯\*, Quan-Hui Liu[1]☯\***

**1** College of Computer Science, Sichuan University, Chengdu, China, **2** Glasgow College, University of Electronic Science and Technology of China, Chengdu, China, **3** Department of Mathematics, North University of China, Taiyuan, China, **4** Complex Systems Research Center, Shanxi University, Taiyuan, China, **5** Laboratory for Computational Epidemiology and Public Health, Department of Epidemiology and Biostatistics, School of Public Health, Indiana University Bloomington, Bloomington, Indiana, United States of America, **6** Lee Kong Chian School of Medicine, Nanyang Technological University, Singapore, Singapore

☯ These authors contributed equally to this work.
\* keisuke.ejima@ntu.edu.sg (K.E.); quanhuiliu@scu.edu.cn (Q.-H.L.)

**Data Availability Statement:** Data and code needed to replicate the results of our analyses are available from GitHub at: https://github.com/QH-Liu/Test-Trace-Isolate.

## Abstract

There are many contrasting results concerning the effectiveness of Test-Trace-Isolate (TTI) strategies in mitigating SARS-CoV-2 spread. To shed light on this debate, we developed a novel static-temporal multiplex network characterizing both the regular (static) and random (temporal) contact patterns of individuals and a SARS-CoV-2 transmission model calibrated with historical COVID-19 epidemiological data. We estimated that the TTI strategy alone could not control the disease spread: assuming $R_0 = 2.5$, the infection attack rate would be reduced by 24.5%. Increased test capacity and improved contact trace efficiency only slightly improved the effectiveness of the TTI. We thus investigated the effectiveness of the TTI strategy when coupled with reactive social distancing policies. Limiting contacts on the temporal contact layer would be insufficient to control an epidemic and contacts on both layers would need to be limited simultaneously. For example, the infection attack rate would be reduced by 68.1% when the reactive distancing policy disconnects 30% and 50% of contacts on static and temporal layers, respectively. Our findings highlight that, to reduce the overall transmission, it is important to limit contacts regardless of their types in addition to identifying infected individuals through contact tracing, given the substantial proportion of asymptomatic and pre-symptomatic SARS-CoV-2 transmission.

## Author summary

Among the various targeted NPIs, the Test-Trace-Isolate (TTI) strategies have been commonly adopted to mitigate and control the spread of infectious diseases. Previous studies evaluating the effectiveness of TTI strategy on SARS-CoV-2 spreading have presented contrasting results, which could be due to the nature of observation studies and imperfect reflection of characteristics of the disease and interventions including TTI in the analyses.

**Funding:** Q.-H. L. has received funding from the National Natural Science Foundation of China (No. 62003230), the National Social Science Foundation of China (No. 20&ZD112). J. L. has received funding from the 111 Project under grant agreement B21044. K. E. has received funding from Lee Kong Chian School of Medicine Start-up Grant (No. NU38OT000297). The funders had no role in study design, data collection and analysis, decision to publish, or preparation of the manuscript.

**Competing interests:** M.A. has received consultancy fees from Seqirus unrelated to this work. All other authors report no competing interests.

To properly account for the contact patterns, here we propose a novel static-temporal multiplex network model describing the static contacts and temporal contacts encountered in different settings (like households, schools, workplaces or general community), which captures the essential nature of social behavior relevant to transmission risk and enables us to assess the effectiveness of the TTI strategy. Our proposed model suggests that TTI strategy solely could not control the spread of SARS-CoV-2 and it should be combined with other social distancing policy to reduce the overall transmission of SARS-CoV-2.

## Introduction

Since the first diagnosed COVID-19 case, the spread of SARS-CoV-2 has caused more than 600 million confirmed cases and 6 million deaths [1] as of 21 December 2022, which has posed overwhelming stress to the public health capacity and presenting huge challenges for the human societies [2]. To control the transmission of SARS-CoV-2 and minimize morbidity and mortality, many countries initially implemented aggressive non-pharmaceutical interventions (NPIs) such as nationwide lockdowns, at a time when effective pharmaceutical treatments were not available [3,4]. However, due to the substantial impact of the lockdowns on the functioning of the society and the availability of new options to limit COVID-19 burden such as vaccines [5], these aggressive NPIs were gradually lifted [6]. Along with vaccination campaigns, milder NPIs (e.g., masks, hand hygiene, and physical distancing) and more targeted NPIs such as reactive school-closures [7–10], screening of individuals in specific occupations [11,12], and Test-Trace-Isolate [13,14] were implemented and became crucial parts of mitigation strategies [15,16].

Among the various targeted NPIs, the Test-Trace-Isolate (TTI) strategy has been widely adopted for many infectious diseases beyond COVID-19, such as SARS [17], H7N2 influenza [18], and Ebola [19]. For COVID-19, TTI yielded contrasting results. Early studies for the UK [20], China [21], South Korea [22] and New Zealand [23] showed that a timely implementation of contact tracing has the potential to contain a SARS-CoV-2, while studies for Canada [24] and France [25] showed the opposite. These discrepancies could be explained by several reasons. First, other interventions were implemented along with TTI, which hinders the possibility to pinpoint the effectiveness of TTI. Second, TTI was implemented differently. For example, regional differences in testing capacity, tracing rate, and population compliance impact the effectiveness of TTI [26]. For example, Contreras et al. found that once the number of new cases exceeds the tracing capacity, a steep rise of new cases is expected to occur in the following weeks [26]. Hellewell et al. showed that, together with the fraction of pre-symptomatic transmission, the delay between symptom onset and isolation had the largest role in determining whether an outbreak was controllable [13]. Chiu et al.'s study indicates that it is necessary to double testing or tracing rate to contain COVID-19 in the US [27]. Davis et al. found that reporting and adherence have remarkable influence on the impact of contact tracing [28]. Third, the mitigation level of TTI is influenced by the characteristic of the disease. Indeed, effectiveness of TTI could be naturally limited by the high proportion of pre-symptomatic and asymptomatic transmission of SARS-CoV-2 [29], because the TTI is primarily triggered by identifying symptomatic patients. Oran et al. [30] estimated that nearly 40% to 45% of SARS-CoV-2 infections occur during the pre-symptomatic and asymptomatic stages, which is crucial to the effectiveness of symptom-driven interventions [31]. Furthermore, a remarkable fraction of transmission from symptomatic individuals occurs before symptom

onset [32–34]. Bi et al. suggested that the effectiveness of isolation and contact tracing highly depends on the number of asymptomatic infections [21]. Thus, studies comprehensively assessing the effectiveness of TTI accounting for the characteristics of the disease and in the presence/absence of other interventions are warranted to define best practices for the use of TTI to control SARS-CoV-2 transmission.

Given that interpersonal contacts are the main transmission route for SARS-CoV-2 and other respiratory pathogens [35][36] and that TTI limits such contacts, contact patterns should be properly accounted in the assessment of the effectiveness of TTI. In fact, fomite transmission of SARS-CoV-2 is considered rare [37,38]. A key factor that limits the effectiveness of TTI is the number of contacts that an index case can remember and ultimately be traced [39]. In general, individual's contacts can be classified into two groups [40–42]. The first group includes individuals that are met regularly, such as household members, schoolmates, and colleagues; here referred to as "static contacts" [43,44]. The second group includes individuals that are met occasionally and/or randomly [45,46], such as individuals met at the supermarket, on the bus, etc. These contacts, here referred to as "temporal contacts", are hard to trace but were proven to be a relevant source of exposure for SARS-CoV-2 [47,48]. Despite their importance, most modeling studies that focused on accessing the impact of TTI on the spread of SARS-CoV-2 have not or only partially reflected such static-temporal structure of the contact network. For example, some studies have not considered individuals' contact networks, and assumed homogeneous mixing either implicitly or explicitly [13,14,26,28,49]. Other studies considered complex contact patterns but connections were static [7,50]. Several studies have attempted to incorporate temporal information in the network structure [51]. One strategy that has been adopted is the use of weighted links to indicate the frequencies or duration of contacts. For example, Firth et al. [52] constructed a static social network based on social contact data and movement data and weighted each contact by the frequency of days that a contact was observed; however, it is worth noting that this type of simplification may limit the capacity of the network model to represent various distributions of epidemic sizes [53]. A second strategy was the use of line graphs to represent the movement from source to target and the corresponding time points. This method requires specific location information and chronological order, which is highly demanding for data acquisition. Because the significance of temporal contact network on pathogen transmission [47,48] and on the different effectiveness of TTI for static (relatively easy to identify) vs. temporal (difficult to identify) contacts, considering both static and temporal contact patterns into a single modeling framework will help us better evaluate effectiveness of TTI.

In this work, we developed a novel infection transmission model considering a static-temporal multiplex network. The transmission model was used to access the effectiveness of TTI on SARS-CoV-2 spread considering an epidemiological context consistent with the initial phase of the COVID-19 pandemic (i.e., ancestral SARS-CoV-2 lineages, fully susceptible population, no vaccines and antiviral therapies). We run epidemic simulations over one year for different values of the reproduction number and under different TTI strategies coupled with interventions restricting contacts. Simulated scenarios are compared in terms of COVID-19 burden (e.g., number of averted infections and deaths) and social and implementation costs of TTI (e.g., peak number of simultaneously isolated individuals, peak number of individuals that needs to be traced).

## Methods

### Overview of transmission simulation with TTI

To simulate SARS-CoV-2 transmission dynamics in a population, we developed a stochastic agent-based model [54] on a static-temporal multiplex network. The disease progression was

modelled through a susceptible-exposed-infectious-removed (SEIR) scheme. The static-temporal multiplex network was composed of two layers: the static contact layer and the temporal contact layer. The links on the static contact layer describe regular contacts, such as contacts among household members, schoolmates, and colleagues, which is assumed to be static over the epidemic period. The links on the temporal contact layer describe occasional contacts, which are assumed to change over time and are updated every day in our simulation. Furthermore, we implemented TTI in the simulation and examined its effectiveness in reducing disease burdens over one year.

## Contact survey data

To reconstruct the network of contacts, we relied on data collected in a contact survey conducted between December 2017 and May 2018 in Shanghai, China [46]. Similar to the POLY-MOD study [55], the survey collected basic socio-demographic information about each study participant (e.g., age, sex) and detailed information about each individual contacted by the participant during the 24 hours period before filling in the questionnaire, where a contact was defined as either a two-way conversation with three or more words in physical presence or a physical skin-to-skin contact. Information collected for each contact included age/age group, sex, contact location and duration, whether the contact was physical or not, and the social setting where the contact took place (e.g., school, workplace). The original contact survey data shows that households, schools, and workplaces were the social settings accounting for the majority of contacts but participants also reported large number of contacts (20+) that were difficult to record individually, like those related to social events (i.e., contacts in the general community).

## Static-temporal multiplex network model

We developed a novel static-temporal multiplex network (**Fig 1A**). The network is composed of two layers: a static contact layer and a temporal contact layer, which represent age-specific "regular" contact patterns (i.e., contacts which occur in households, schools, and workplaces) and "occasional" contact patterns (e.g., contacts taking place during social events), respectively. In each layer, nodes and links represent individuals and contacts, respectively. A population of 100,000 individuals is synthesized following the age distribution from the census data of Shanghai, China [56] and connected on the static contact layer and the temporal contact layer. A configuration model was used to construct both the static and temporal contact layers [57], preserving the age-specific regular and occasional contact patterns observed in the contact survey conducted in Shanghai, China [46]. On the static contact layer, links are generated following age-dependent contact frequencies (**Fig 1B**). Specifically, for each node, the number of links was sampled from the corresponding age-specific distribution. The static contact layer is assumed constant (thus the links are not re-generated or disappear during the simulation of an epidemic). On the temporal contact layer, social events are generated, in which all participants are connected. Social events are generated following the proportion of individuals participating to social events as derived from the contact survey data (**Fig 1C**), the distribution of the event size (**Fig 1D**), and age distributions of the participants of the social event (**S3 and S4 Figs**). Notably, school-age or working-age individuals are more likely to have contacts on the temporal contact layer (**Fig 1C**). The links on the temporal contact layer are updated at each time step of the simulation (i.e., every day). The average numbers of contacts for the static and temporal contact layers are 7.15 and 13.7, respectively, which add up to 20.85 contacts per day and is consistent with the original data (an average of 19.3 contacts per day) and other estimates obtained before the COVID-19 pandemic in European countries (e.g., 19.77 and 17.46

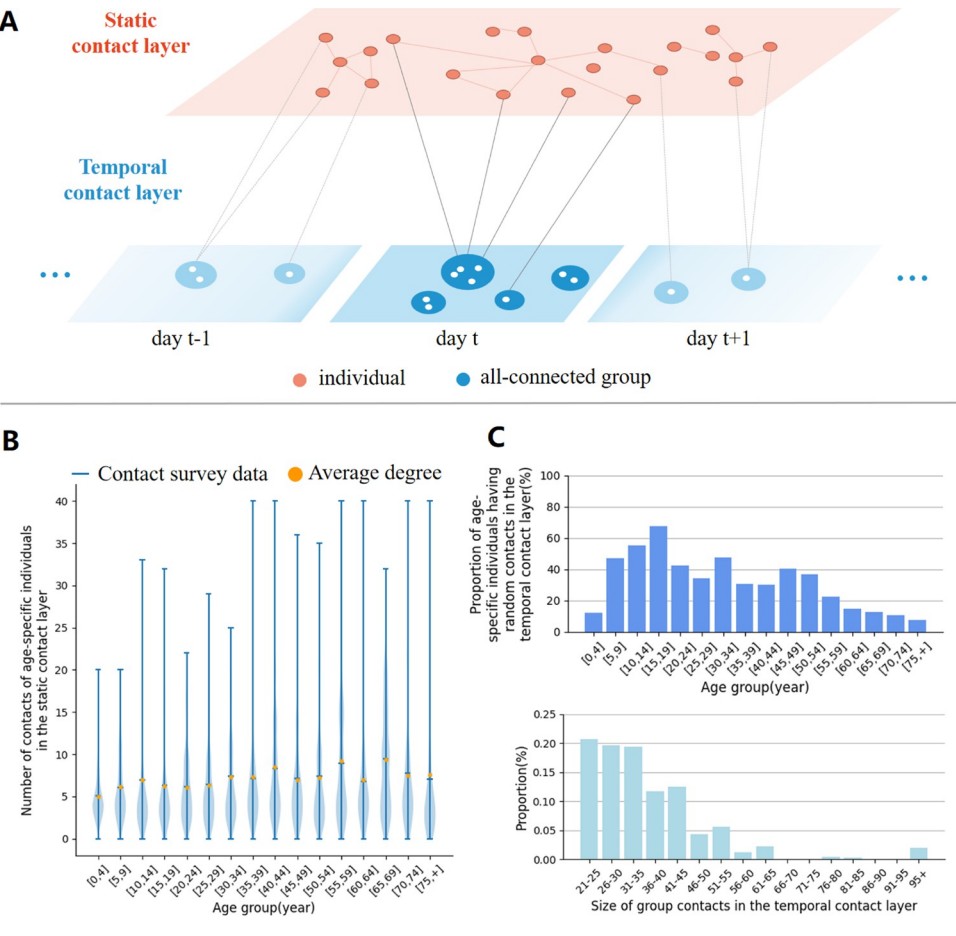

**Fig 1. Static-temporal multiplex network model and social contact pattern. A** A schematic illustration of the static-temporal multiplex network, which is composed of two layers: static contact layer and temporal contact layer. The links on the static contact layer represent regular interactions occurring in households, schools, and workplaces, whereas those on the temporal contact layer represent occasional contacts occur in gatherings and events. The links on the static contact layer are constant, whereas the links on the temporal layer are updated every day. **B** The age-specific contact frequency from the survey and the synthesized static layer. **C** The age-specific proportion of individuals joining gathering and events. **D** The size distribution of gathering and events.

contacts per day on average were reported in Italy and Luxembourg, respectively) [55]. More details are available in **S1 Supplementary** **Methods** (1. Contact survey data for the static-temporal multiplex network and 2. Static-Temporal Multiplex Network).

## SARS-CoV-2 transmission model

We developed a stochastic agent-based model of SARS-CoV-2 transmission to evaluate the impact of the test-trace-isolate strategy in mitigating COVID-19 burden. Briefly, SARS-CoV-2 transmission is simulated according to an SLIR (susceptible, latent, infectious, removed) scheme, where infectious individuals were further divided into pre-symptomatic (P), symptomatic (S), and asymptomatic (A) individuals (**S9 Fig**). If a susceptible individual i is connected with an infectious (either pre-symptomatic, symptomatic, or asymptomatic) individual j, susceptible individual i could be infected by infectious individual j with a layer-specific probability and become latent. After a latent period, latent individuals either become infectiousness pre-symptomatic or asymptomatic. Once they develop symptoms, pre-symptomatic

individuals are categorized as symptomatic. Finally, when symptomatic and asymptomatic individuals are no longer infectious, they are categorized as removed. Given that we are interested in simulating a short time period (365 days), removed individuals are assumed to be fully protected from future re-infections.

The probability of infection for each pair of infected and susceptible individuals is determined by three factors: age-specific susceptibility to infection [33], type of contact (static or temporal) [32], and the status of the infectious individual (P, I or A) [33]. The incubation period was assumed to follow a Gamma distribution with a mean of 6.3 days and a standard deviation of 4.3 (shape = 2.08, rate = 0.33) [33]. We considered transmission to start 2 days before symptom onset [33]. The duration of the infectious period was chosen such that distribution of the generation time in the simulation match an estimated mean of 7.0 days (IQR: 3.6–11.3) [32].

As a measure of disease burden, we estimated the cumulative number of infections, symptomatic infections, hospitalizations, ICUs, and deaths from the simulated epidemics. As a measure of both the social and implementation cost of TTI, we estimated the peak daily number of individuals that need to be traced (i.e., those with positive test result) and the peak daily number of individuals that are simultaneously isolated.

In our baseline analysis, we considered $R_0 = 2.5$ (thus, if not specified, $R_0 = 2.5$). Simulations were initialized with a fully susceptible population (thus reflecting the situation at the start of the pandemic), except for 5 randomly selected latent individuals. For each analysis, results are based on 100 stochastic model realizations. A complete list of model parameters is reported in **S1 Table**. A more detailed description of the transmission model is available in **S1 Supplementary Methods** (3. SARS-CoV-2 Transmission model).

## Test-Trace-Isolate strategy

Symptom-based testing, contact tracing, and case isolation (namely, the Test-Isolate-Trace strategy) was implemented as follows:

1. **Symptom-based testing.** A fraction (80% in the baseline) of individuals who develop symptoms are tested by reverse transcription polymerase chain reaction (RT-PCR) tests. The time intervals between symptom onset and sample collection, and between sample collection and laboratory diagnosis were accounted for as well as the sensitivity of the RT-PCR test, which depends on the timing of sample collection [58].

2. **Contact tracing.** Contact tracing is triggered by a positive test result (see an example in **Fig 2**). All contacts of each positive individual on the static contact layer are traced. On the temporal contact layer, those who are connected within 4 days before sample collection are traced with a probability of 50%. The tracing process is assumed to be completed when a positive individual is identified, and the sample is immediately collected from the traced individuals.

3. **Isolation.** Before sample collection, links on both layers are active as usual. Between sample collection and test result, links on temporal contact layer are disconnected, as we assume that individuals would avoid unnecessary activities (**Fig 2**). If the test result is negative (false negative), the disconnected links recover, as individuals go back to normal activity. If the test result is positive (true positive), the links of infected individuals on both layers are disconnected (i.e., isolated) for 14 days.

Full description of the TTI strategy is available in **S1 Supplementary Methods** (5. The Test-Isolate-Trace strategy), and values and description of all TTI parameters are listed in **S2 Table**.

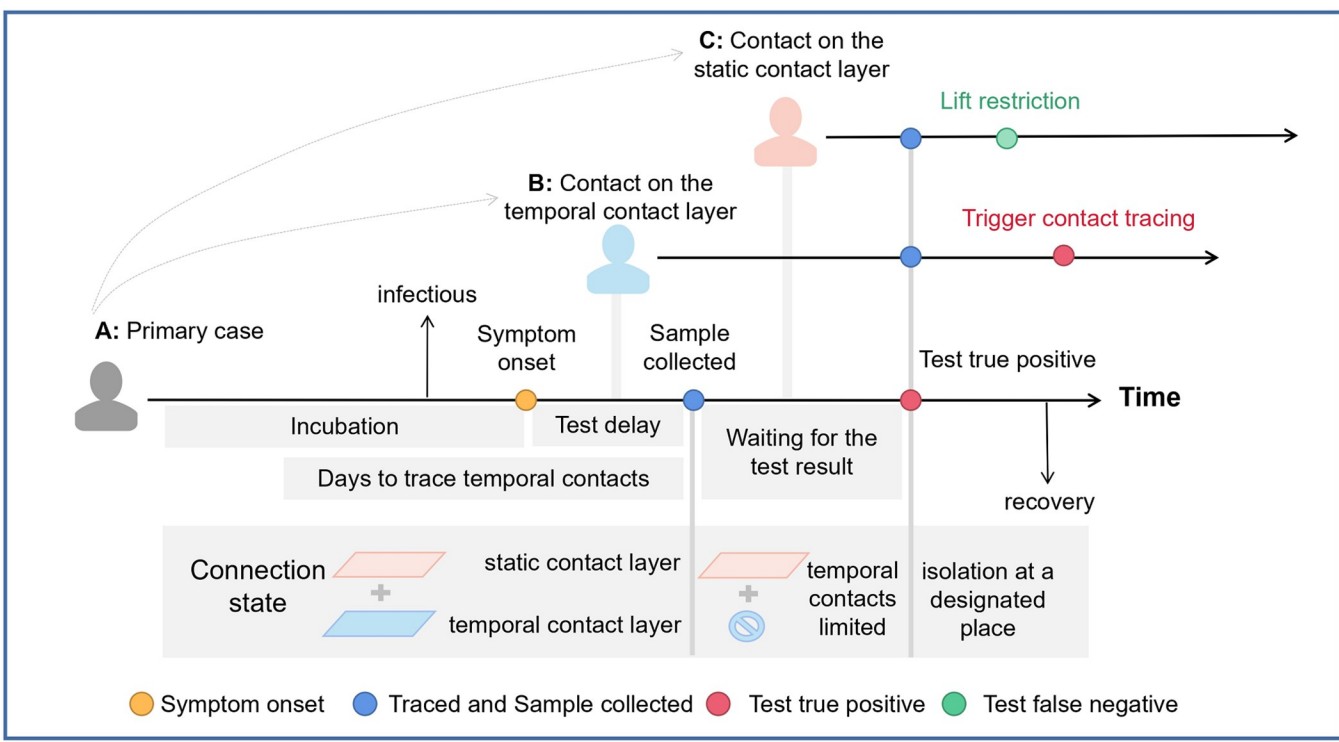

**Fig 2. Example of the Test-Trace-Isolate strategy.** Once the primary case **A** develops symptoms, the sample is collected with delay. After the sample collection, **A**'s contacts on the temporal contact layer are disconnected until the test result is returned. If the test result is positive (which is the case in this figure), **A** is isolated, thus the contacts on both layers are disconnected over 14 days. If the test result is negative (i.e., false negative), the disconnected links recover. **A** infects individual **B** through a contact on the temporal contact layer (before **A**'s sample collection) and infects individual **C** through a contact on the static contact layer before the test result is returned. Individuals **B** and **C** are traced and tested immediately after the infection of **A** is confirmed. The links of **B** and **C** on the temporal layer are disconnected until the test results are returned. If the infection of the traced individuals (**B** in this figure) is confirmed, further contact trace is triggered. If the infection of the traced individuals (**C** in this figure) is not confirmed (i.e., false negative), the disconnected links recover.

## Results

### Effectiveness and social/implementation cost of TTI

Without interventions, the cumulative number of infections (i.e., infection attack rate; IAR) and deaths reached 81.2% (95% CI: 80.8–81.7%) (**Fig 3A**) and 4.5 (95% CI: 4.2–4.9) per 1,000 people (**Fig 3B**), respectively. The proportion of pre-symptomatic transmission was 57.0% (95% CI: 56.3–57.7%), which is similar to the value reported in the analysis of the transmission events in Hunan, China [33]. The Test-Trace-Isolate strategy reduced infections and deaths by 24.5% (95% CI: 23.2–26.4%) and 19.3% (95% CI: 8.7–28.8%), respectively, which is consistent with previous studies [26]. The results for other metrics of disease burden are reported in **S1 Supplementary Methods** (**S10 Fig**). The peak daily number of traced individuals and the peak daily number of simultaneously isolated individuals were 13.7 (95% CI: 13.0–14.5) per 1,000 people (**Fig 3C**), and 177.3 (95% CI: 171.4–184.0) per 1,000 people (**Fig 3D**), respectively.

We run a variety of sensitivity analyses varying the parameters regulating TTI. Two parameters, the probability that contacts on the temporal layer to be successfully traced and the number of days contacts are traced on the temporal contact layer before sample collection of the primary case, are positively associated with a reduction of IAR (**Fig 4A and 4B**), as more infected individuals are isolated. A shorter time interval from symptom onset to sample collection is associated with a higher reduction of IAR (**Fig 4C**), because contacts of tested

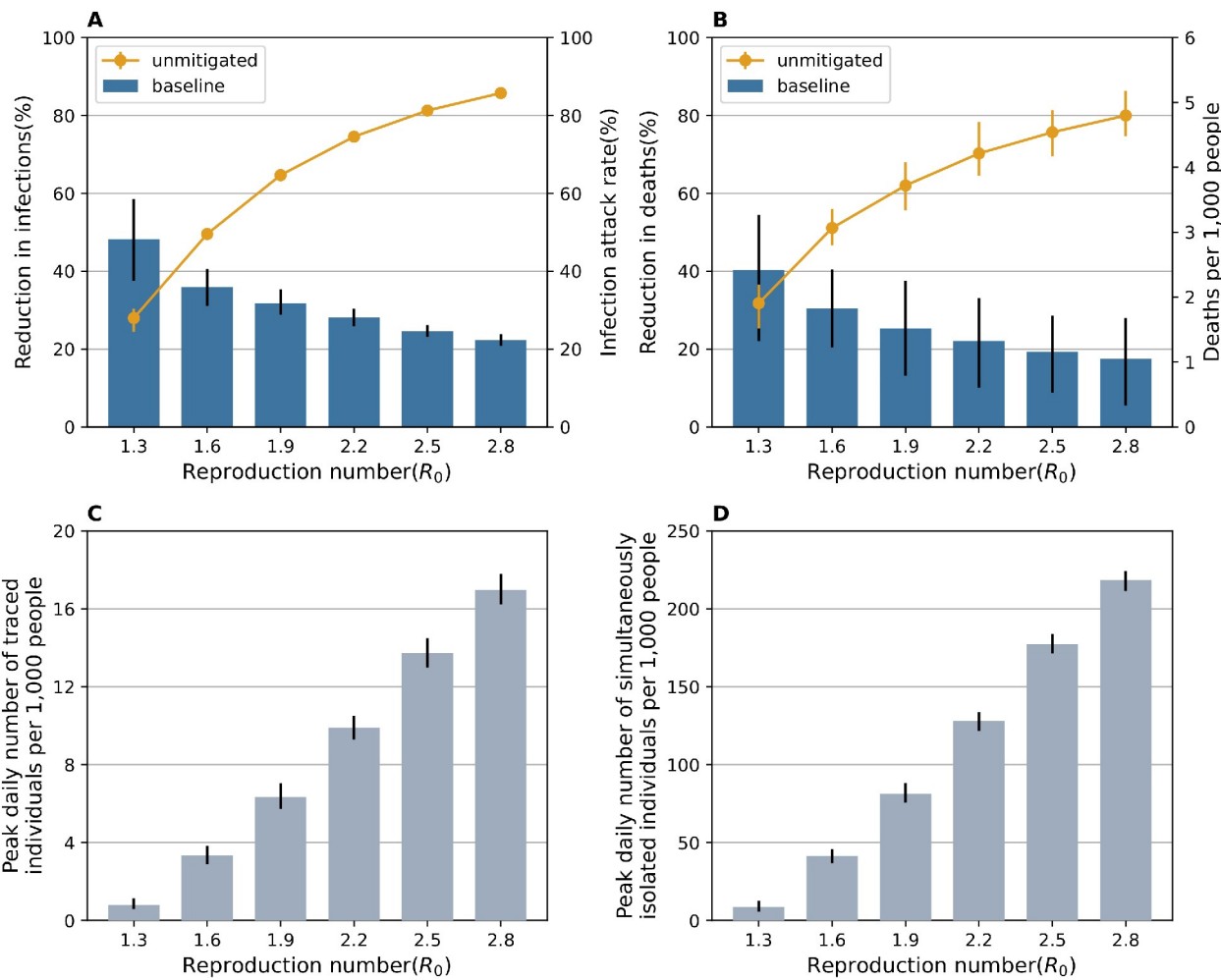

**Fig 3. Effectiveness and social/implementation cost of TTI. A** Infection attack rate (IAR) without TTI strategy (unmitigated) and reduction in the cumulative infections by the Test-Trace-Isolate strategy (baseline) with different reproduction number $R_0$. The vertical error bars indicate the 95% CI. **B** as in **A**, but for the number of deaths per 1,000 people and reduction in deaths. **C** Peak daily number of traced individuals with positive test results (thus contact tracing is needed) per 1,000 people. **D** Peak daily number of simultaneously isolated individuals per 1,000 people.

individuals on the temporal contact layer are promptly disconnected from the network after the sample collection, in agreement in previous studies [7].

The impact of the time interval from sample collection to laboratory diagnosis ($T_{cr}$) on the reduction of IAR depends on the pathogen transmissibility ($R_0$) (**Fig 4D**). Longer $T_{cr}$ decreases the IAR reduction under a low transmissibility scenario ($R_0 = 1.3$); however, it increases the IAR reduction under a high transmissibility scenario ($R_0 = 2.5$). This complex association is due to the balance between the probability of secondary transmission from isolated individuals and the probability that secondary infections can be effectively isolated. The former probability decreases with shorter $T_{cr}$, because their links on the static layer remain active while they wait for the test result. Furthermore, it increases with high transmissibility. Meanwhile, the probability that secondary infections can be effectively isolated decreases with shorter $T_{cr}$, under which the traced (and infected) individuals may not be properly identified by the test because of low test sensitivity during the early phase of infection (many secondary transmissions occur around symptom onset of primary cases, given that the incubation period [6.3 days] and the generation time [7.0 days] are close). Furthermore, it decreases with high transmissibility

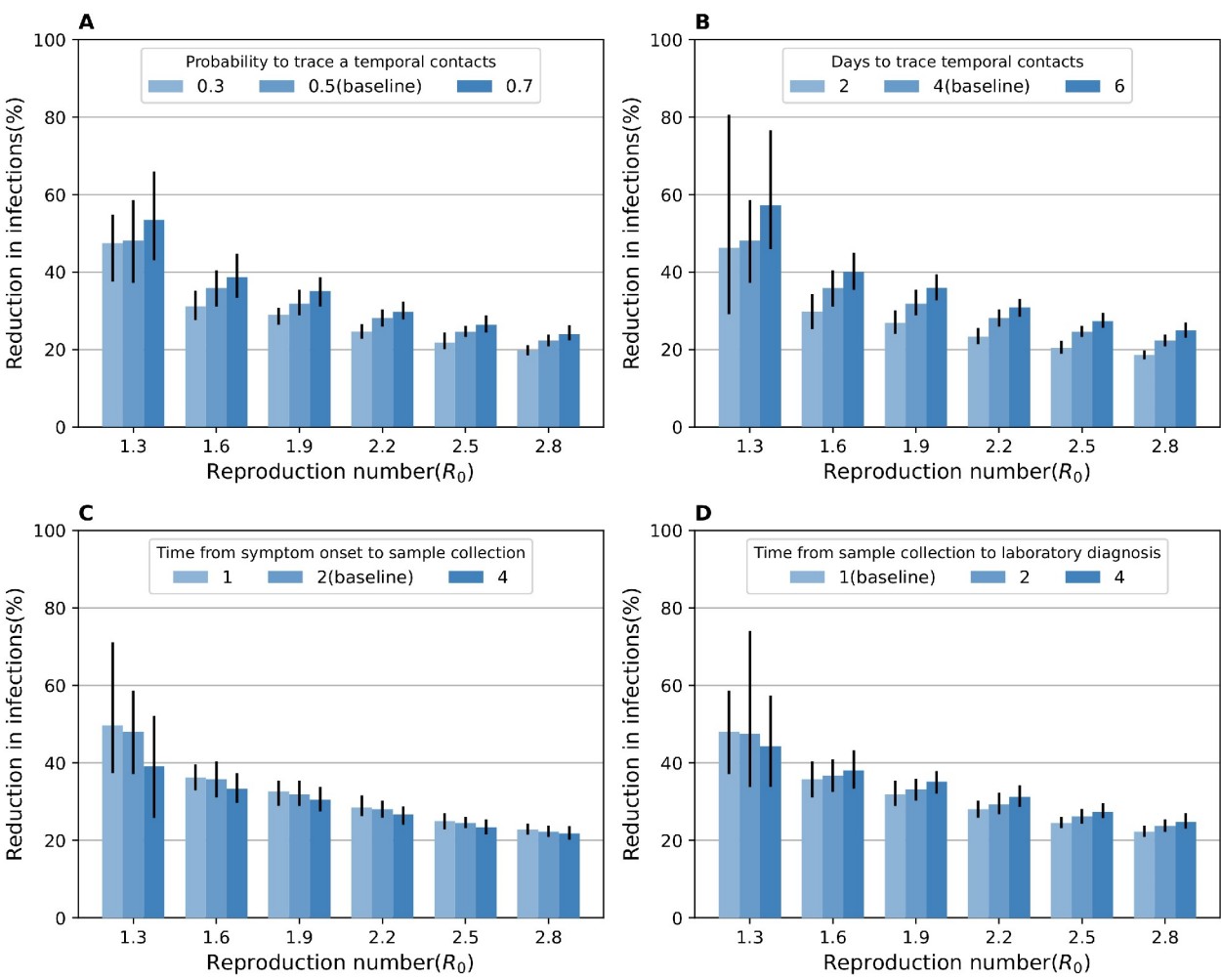

**Fig 4. Sensitivity analyses varying the parameters regulating the test and trace processes. A** Reductions in the cumulative infections with different probabilities that contacts on the temporal layer to be successfully traced. The vertical error bars indicate the 95% CI. **B** as in **A**, but the days to trace contacts on the temporal contact layer before sample collection of the primary cases were varied. **C** as in **A**, but the time from symptom onset to sample collection was varied. **D** as in **A**, but the time from sample collection to laboratory diagnosis was varied.

because more asymptomatic or pre-symptomatic individuals (lower sensitivity) are tested. Indeed, the observed true positive rate, especially among those identified through contact tracing, was reduced by the high value of $R_0$ (**S11B and S11D** Figs). To quantify the impact of this higher test positive rate among traced individuals, we designed a null model with the same parameter settings except for the test sensitivity for the pre-symptomatic and asymptomatic individuals, which was set so that the observed true positive rate among traced individuals is the same as for the model with $T_{cr} = 4$ (**Fig 5B**). Thus, the difference in IAR reduction between the null and baseline models (27.6% with $R_0 = 1.3$; **Fig 5A**) is due to the increased true positive rate for traced individuals. This difference between the null model and the model with $T_{cr} = 4$ (23.9% with $R_0 = 1.3$; **Fig 5A**) is associated with a larger number infections occurring while test results are not available yet. Thus, when $R_0$ is low (i.e., 1.3), IAR is reduced by 3.7%, while it increases by 5.2% when $R_0$ is high (2.5).

Overall, from our sensitivity analyses, we found that TTI can be implemented more efficiently by increasing the likelihood of tracing contacts or reducing testing delays. However, TTI alone is not sufficient to control an epidemic even in a best-case scenario. To mitigate COVID-

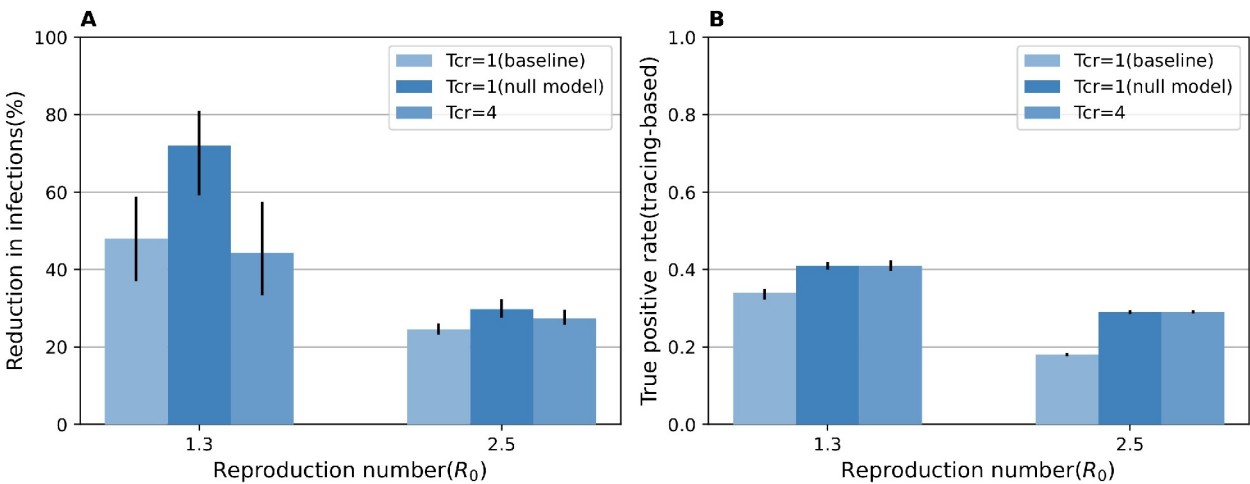

**Fig 5. Analysis of baseline with longer delay varying the time from sample collection to laboratory diagnosis. A** Reduction in the cumulative infections. **B** True positive rate among the traced individuals. "$T_{cr} = 1$ (baseline)" and "$T_{cr} = 4$" are the models setting the time from sample collection to laboratory diagnosis as 1 day or 4 days, respectively. "$T_{cr} = 1$ (null model)" is the model setting the time from sample collection to laboratory diagnosis as 1 day, but the true positive rate among the traced individuals is the same as the model with "$T_{cr} = 4$". Other parameter settings are the same.

19 burden, combining TTI with other interventions would need to be considered. Indeed, during the COVID-19 pandemic, multiple non-pharmaceutical interventions (NPIs), such as mask wearing and improved hygiene practices [4] were implemented along with TTI, the combination of which resulted in a reduction of the transmission risk (and of the effective reproduction number). Here we run simulations for different values of the reproduction number to resemble the effect of different NPIs that are implemented to lower the transmission risk. In addition, we also run simulations with a higher value $R_0$ (i.e., = 2.8) to account for the variability in $R_0$ estimates, for instance due to different variants, socio-demographic characteristics, and contact patterns [13]. Our simulations show that TTI is more effective for lower values of $R_0$. For example, when $R_0$ is 1.3, TTI can reduce the total number of infections and deaths by 48.1% (95% CI: 37.6–58.5%) and 40.4% (95% CI: 23.5–54.4%), respectively. The social and implementation costs of TTI are reduced for lower $R_0$ values. For example, when $R_0$ is 1.3, the peak daily number of individuals to be traced and the peak daily number of simultaneously isolated individuals drop to 0.8 (95% CI: 0.6–1.1) per 1,000 people and 8.8 (95% CI: 5.7–12.5) per 1,000 people, respectively. Therefore, combining TTI with NPIs not only increases the effectiveness of TTI strategy, but also reduces the social and implementation costs associated with TTI.

We further run sensitivity analyses varying the parameters unrelated to TTI, namely: (1) infectiousness of asymptomatic individuals, $\chi(A)$; (2) the number of initially infected individuals; (3) the probability that a symptomatic individual is tested, $P_{test}$. The number of initially infected individuals had no impact on the effectiveness of TTI (**S13 Fig**); however, lower $\chi(A)$ and higher $P_{test}$ were associated with higher effectiveness of TTI (**S12A and S14A Figs**) and deaths (**S12B and S14B Figs**).

A point of discussion during the pandemic has been whether individuals should be isolated while waiting for the result of a test [59–61]. In the main analysis, we assumed that only the links on the temporal contact layer are disconnected while waiting for the test result. Here, we run two sensitivity analyses: (1) links on both layers are disconnected (TTI-strict) or (2) all links remain active (TTI-relaxed) while waiting for the test result. The effectiveness of TTI was lower when a less restrictive TTI is considered (**Fig 6A**). Specifically, for $R_0 = 2.5$, the reduction of the IAR was reduced by 45.3% and increased by 17.6% when considering TTI-relaxed and

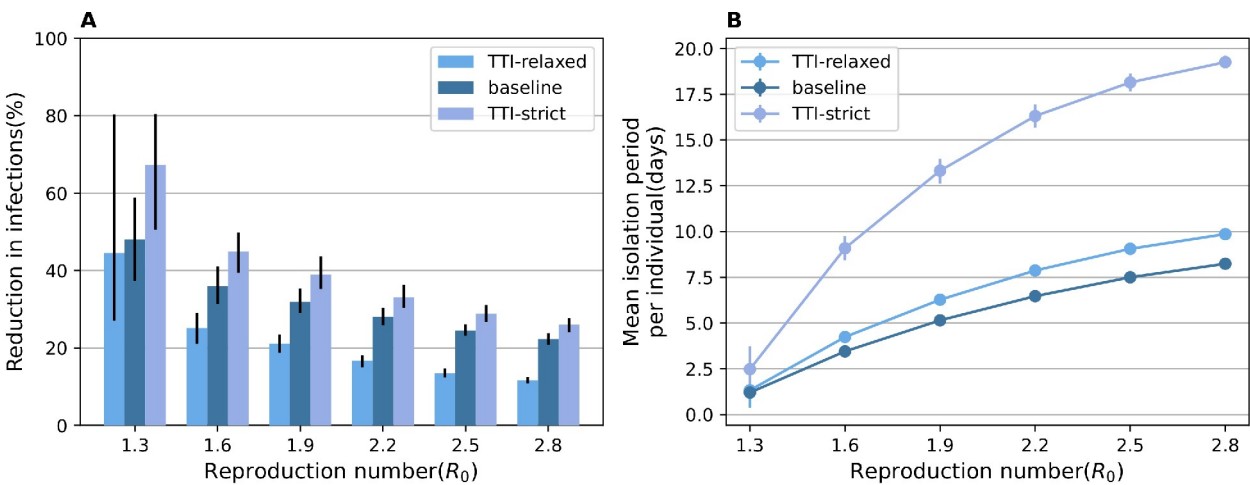

**Fig 6. Effectiveness and burden of TTI-relaxed and TTI-strict. A** Reductions in the cumulative infections with different TTI strategies. (TTI-relaxed) Contacts on both layers remains while the test results are waited. (TTI-strict) Contacts on both layers are disconnected while the test results are waited. **B** as in **A**, but for the mean isolation period.

TTI-strict, respectively (**Fig 6A**), as compared with TTI (baseline). Meanwhile, the mean isolation period was 1.5 days and 10.5 days longer for TTI-relaxed and TTI-strict, respectively (**Fig 6B**). The longer isolation period of TTI-relaxed was due to the larger number of infections, whereas that of TTI-strict is due to its mechanism, as individuals are isolated while waiting for the test result. These findings suggest that unnecessary gatherings and social activities should be avoided while waiting for test results; however, avoiding regular activities (such as going to schools or offices) may not be necessary, considering the trade-offs between the effectiveness of the TTI strategy and its social and implementation costs.

## Reactive distancing policy

We considered to combine TTI with the other interventions by varying $R_0$ in the sensitivity analysis. However, reduced values of $R_0$ does not have a practical interpretation as there is no clear connection between the magnitude of reduction in $R_0$ and specific interventions. Here, we explicitly simulated a reactive social distancing policy as an example. Limiting social events and gathering has been adopted in many countries and regions as a social distancing policy to limit SARS-CoV-2 transmission, especially at early phase of the pandemic.

We analyzed two social distancing policies: 1. reactive social distancing, i.e., limiting unnecessary gathering and events only (a fraction of contacts on temporal contact layer is disconnected) [62], 2. reactive "all-level" distancing, limiting both unnecessary gathering and events and necessary activities (a fraction of contacts on both contact layers is disconnected) [63]. Those policies are triggered when the cumulative number of detected cases exceeds a predefined threshold value. Details are reported in **S1 Supplementary Methods** (6. Reactive distancing policy) and **S3 Table**.

We estimate reactive social distancing to have a substantial impact in reducing COVID-19 burden (**Fig 7A and 7B**). For example, when $R_0 = 2.5$ and 50% or 75% of the contacts on the temporal layers are disconnected ($P_{social} = 0.5$ or 0.75), the IAR reduction reached 37.8% (95% CI: 36.4–39.7%) and 45.6% (95% CI: 44.4–46.7%), respectively. Reactive "all-level" distancing further reduced COVID-19 burden (**Fig 7C and 7D**). When 30% of contacts on the static layer and 50% of contacts on the temporal layer are disconnected ($P_{social} = 0.5$, $P_{static} = 0.3$), 68.1% (95% CI: 64.3–71.3%) of infections could be averted (**Fig 7C**).

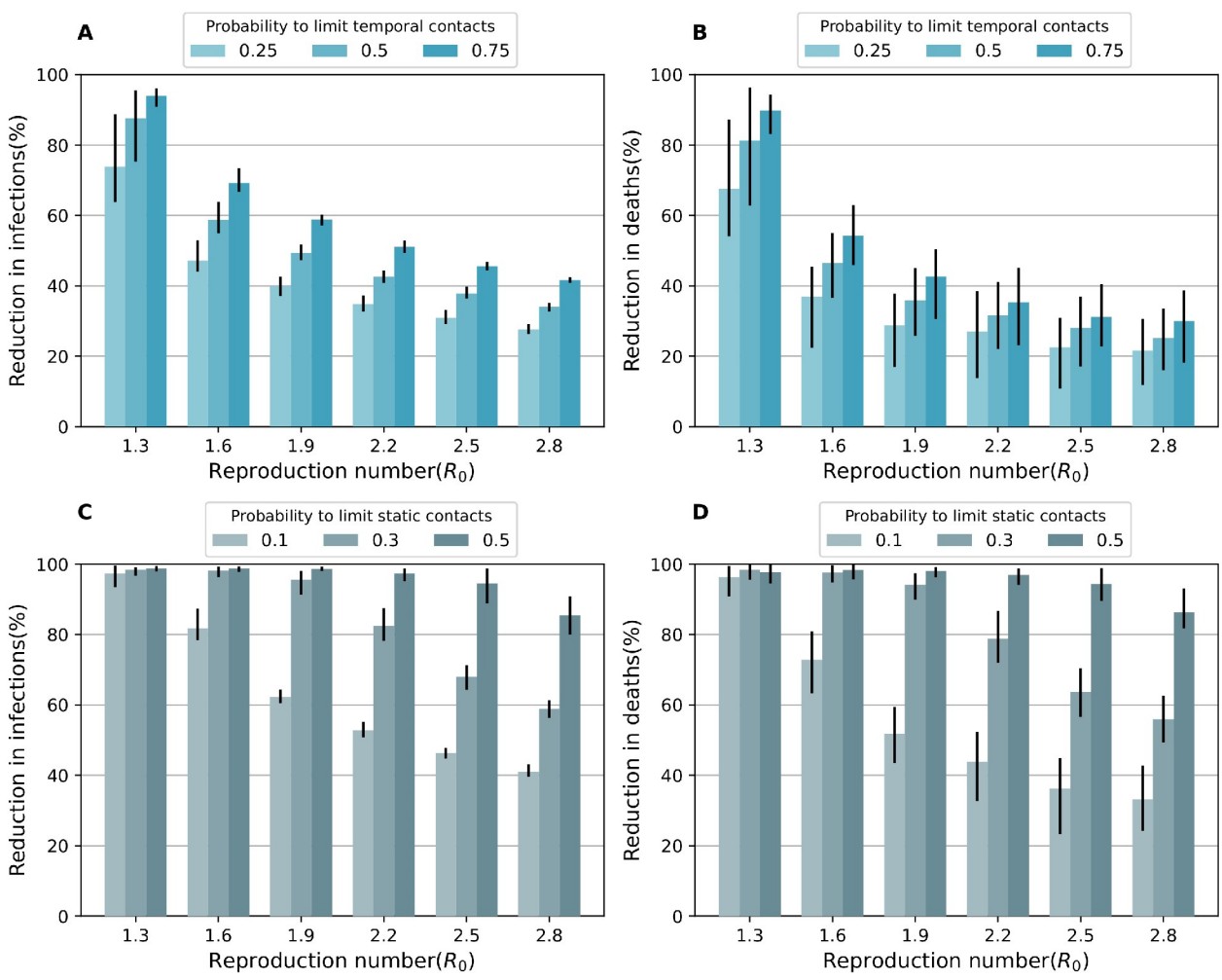

**Fig 7. Effectiveness of reactive distancing policy coupled with the TTI strategy. A** and **B** Reactive social distancing: a fraction of contacts on temporal contact layer are disconnected. **A** for the reduction in the cumulative infections, **B** as in **A**, but for the reduction in deaths. **C** and **D** Reactive all-level distancing: 50% of contacts on temporal layer are disconnected and a fraction of contacts on static contact layer is disconnected. **C** for the reduction in the cumulative infections, **D** as in **C**, but for the reduction in deaths. Those policies are triggered when the cumulative number of detected cases exceeds 50.

## Discussion

Test-Trace-Isolate strategies have been commonly adopted to mitigate and control the spread of infectious diseases [64]. Previous modelling studies have identified several key factors that are crucial to determine the effectiveness of TTI, such as the reproduction number and the proportion of transmissions that occur before symptom onset [13].

On top of those previous studies, we incorporated an essential feature of social behavior shaping the transmission risk. TTI primarily serves to control (in-person) contacts between individuals. However, contacts are highly heterogeneous, and incorporating proper contact patterns in transmission models is essential to assess TTI strategies. Here, we proposed a novel static-temporal multiplex network that incorporates two complementary types of contacts: regular contacts on a static contact layer and random contacts on temporal contact layer. On this comprehensive contact network, we developed a stochastic agent-based SARS-CoV-2 transmission model, which enabled us to implement individual-level interventions accounting for the natural heterogeneity of human contact patterns.

Using our model, we provided a quantitative assessment of the effectiveness of TTI strategies, and found that TTI alone can only partially mitigate COVID-19 burden. This is due to the high proportion of pre-asymptomatic and asymptomatic transmission, which hinders the successful and prompt identification of infected individuals when a symptom-based trigger is considered. When $R_0$ is 2.2, given that infectiousness onset almost coincides with symptom onset (infectiousness appears 0.51–0.77 days before symptom onset on average), Peak et al [65]. found that tracing contacts and quarantine or active monitoring individuals triggered by symptom presence have a certain potential to control the SARS-CoV-2 spreading. However, in situations where less resources are available for tracing, or the adherence to such interventions is low, the effectiveness of TTI could be low. For example, if less than 20% of the population follows the rule of self-reporting and isolation due to symptoms or tracing, large outbreaks are unlikely to be avoided even with scaled-up tracing and quick turnaround testing [28]. We further examined reactive social distancing strategies to be combined with TTI. Specifically, we considered two types of reactive social distancing: 1) a moderate one, which limits only non-essential contacts (e.g., gatherings, social events), thus reducing contacts on the temporal contact layer only; 2) and a strict one, which limits not only non-essential contacts but also a part of regular contacts on the static contact layer (similar to what happens when reactive school and workplace closures are adopted). We found that limiting temporal contacts alone could avert less than 50% of the infections, especially when the transmissibility potential is high ($R_0 \geq 2.2$). Furthermore, limiting both temporal and regular contacts could remarkably enhance the effectiveness of the TTI strategies, as the main transmission route of SARS-CoV-2 transmission is through regular contacts. Thus, especially when effective therapeutics or vaccines were not available, stay-at-home orders to suppress epidemic spread were considered.

Our study has several strengths. First, our model is calibrated using epidemiological and contact survey data. In particular, the structure of the contact network is based on individual-level contact patterns data stratified by age and social setting. However, it is worth noting that, to reconstruct the static-temporal network, we used contact survey data collected before the COVID-19 pandemic. The pandemic has altered the contact patterns of the population in ways that go beyond the implementation of social distancing policies, including individual considerations about the perceived risk of infection [66]. Future studies are needed to estimate contact patterns in the post-pandemic era. Second, we estimated the social and implementation costs of TTI as well as its effectiveness. The social and implementation costs of TTI can be defined in different ways, but further cost-effectiveness analyses are needed to estimate economic costs and broader social implications of TTI strategies as well as other interventions. Finally, our stochastic agent-based model with its static-temporal network structure allows the analysis of individual-level interventions with a high degree of fidelity and flexibility. In this study, we analyzed TTI; however, the model can be used to examine other individual-level NPIs as well as pharmacological interventions, such as vaccination programs, and antiviral treatments and prophylaxis.

Our study has several limitations. First, we grouped contacts that take place in the household, school, and workplace settings into a single layer (i.e., static contact). Therefore, our network does not explicitly replicate the structure of each household, school, and workplace. As such, our network cannot capture the saturation effect of household transmission observed in previous studies [67,68], which may lead our model to slightly overestimate the infection attack rate. Moreover, we did not consider heterogeneity in the transmission risk for contacts on the same layer. For example, the transmission risk of contacts between household members may differ from that of contacts between classmates, and transmission may also differ between individuals due to different behaviors. Further studies are needed to investigate the granularity of contact patterns that is needed to capture epidemiological dynamics with/without

interventions. Second, in our simulations, we assumed complete connection among people attending the same social event, which might not reflect the reality. Contacts may also depend on types of individual (not just age, as considered in this study). The assumption of full connectivity may increase the risk of pathogen transmission among individuals in the same group, as each susceptible individual in that group can be infected by any infectious member of the group. Third, our simulations assumed an instantaneous turnaround time for contact tracing, but contact tracing is often implemented through interviews conducted by an operator; thus, it may take several days to gather all information and identify contacts. Kretzschmar et al. 's study [69] have shown that the tracing delay has certain impact on the contact tracing effectiveness, although lower than the testing delay. Thus, the assumption of an instantaneous turnaround time for contact tracing may lead to a slightly overestimation of the effectiveness of TTI. Fourth, we did not consider the possibility of false positive test results. Although rare, false positive results can increase the social and implementation costs of TTI. This aspect can be particularly relevant for cost-effectiveness analyses.

## Conclusion

In conclusion, we proposed a novel static-temporal multiplex network that captures contact patterns. Using this model, we examined the effectiveness and the social/implementation costs of TTI and identify key parameters regulating its effectiveness. We found that TTI alone could not control a COVID-19 outbreak. The infection attack rate can however be reduced by 24.5% when $R_0 = 2.5$; when NPIs are implemented to lower the transmission risk (e.g., $R_0 = 1.3$), about 48.3% of the infections can be averted. Our findings highlight the importance of combining TTI with an overall reduction of transmission obtained by limiting contacts regardless of their types, given the substantial proportion of asymptomatic and pre-symptomatic transmission and high reproduction number of the ancestral SARS-CoV-2 lineages as well as subsequent variants.

## Supporting information

**S1 Supplementary Methods. 1. Contact survey data for the static-temporal multiplex network.** 2. Static-Temporal Multiplex Network. 3. SARS-CoV-2 Transmission model. 4. Calibration. 5. The Test-Trace-Isolate strategy. 6. Reactive distancing policy.
(DOCX)

**S1 Fig. The age-group-dependent contact frequency of individual contacts from 0 to 39.**
(TIF)

**S2 Fig. The age-group-dependent contact frequency of individual contacts over 40.**
(TIF)

**S3 Fig. The Gaussian kernel fitted age-group-dependent age distribution of contactees of individual contacts from 0 to 39.**
(TIF)

**S4 Fig. The Gaussian kernel fitted age-group-dependent age distribution of contactees of individual contacts over 40.**
(TIF)

**S5 Fig. The Gaussian kernel fitted age-group-dependent age distribution of participants in gathering and events from 0 to 39.**
(TIF)

**S6 Fig. The Gaussian kernel fitted age-group-dependent age distribution of participants in gathering and events over 40.**
(TIF)

**S7 Fig. Contact matrix representing the static contact layer.** Colors represent the mean number of contacts between an individual in given age group with individuals with any age group.
(TIF)

**S8 Fig. Contact matrix representing the temporal contact layer.** The panels show two realizations of the temporal contact layers for two randomly selected day of a simulation. Colors represent the mean number of contacts between an individual in given age group with individuals with any age group.
(TIF)

**S9 Fig. SARS-CoV-2 transmission model.** The detailed description of symbols in the figure referred to **S1 Table.**
(TIF)

**S10 Fig. Disease burdens under the baseline TTI strategy. A.** The number of symptomatic infections per 1,000 people without the TTI strategy (orange dots) and the reduction by the baseline TTI strategy (blue bars). Reproduction numbers ($R_0$) are varied. The vertical error bars indicate the 95% CI. **B.** The same as **A**, but using the number of hospitalization as the disease burden. **C.** The same as **A**, but using the number of ICU patients as the disease burden.
(TIF)

**S11 Fig. Impact of the time from sample collection to laboratory diagnosis. A.** Average number of secondary infection (per a primary case) while waiting for the test results. **B.** observed true positive rate. **C.** as in **B**, but only for the symptomatic individuals. **D.** as in **B**, but only for those recruited through contact tracing.
(TIF)

**S12 Fig. Impact of the relative infectiousness of asymptomatic individuals. A.** Reduction in the cumulative infections under different reproduction number and the relative infectiousness of asymptomatic cases. The vertical error bars indicate the 95% CI. **B.** as in **A**, but reduction in deaths.
(TIF)

**S13 Fig. Impact of the number of initial infected individuals.** A. Reduction in the cumulative infections under different reproduction number and the initial number of cases. The vertical error bars indicate the 95% CI. **B.** as in **A**, but reduction in deaths.
(TIF)

**S14 Fig. Impact of the probability to test a symptomatic individual. A.** Reduction in the cumulative infections under different reproduction number and the probability to test a symptomatic individual. The vertical error bars indicate the 95% CI. **B.** as in **A**, but reduction in deaths.
(TIF)

**S15 Fig. Impact of the reactive threshold values. A.** Reduction in the cumulative infections under different reproduction number and the reactive threshold values for the reactive social distancing. The vertical error bars indicate the 95% CI. **B.** as in **A**, but reduction in deaths. **C.**

as in **A**, but for the reactive all-level distancing. **D.** as in **C**, but reduction in deaths.
(TIF)

**S16 Fig. Daily new infections and new deaths.** A. and **B.** for the daily new infections and daily new deaths under the unmitigated scenario (without TTI) with reproduction number of $R_0 = 1.3, 2.5$ and $2.8$, the shadows indicate the 95% CI; **C.** and **D.** for baseline where TTI is implemented under the scenario that the reactive social distancing is triggered when the cumulative number of detected cases exceeds 50, specifically, 50% of contacts on the temporal layer are disconnected, and under the scenario that the reactive all-level distancing is triggered when the cumulative number of detected cases exceeds 50, specifically, 50% of contacts on the temporal layer are disconnected and 30% of contacts on the static contact layer are disconnected. $R_0 = 2.5$. **E.** and **F.** as in **C.** and **D.**, but $R_0 = 2.8$.
(TIF)

**S1 Table. Parameters for SARS-Cov-2 transmission model and disease burden.**
(DOCX)

**S2 Table. Parameters for TTI.**
(DOCX)

**S3 Table. Parameters for reactive distancing policies.**
(DOCX)

## Author Contributions

**Conceptualization:** Marco Ajelli, Keisuke Ejima, Quan-Hui Liu.

**Data curation:** Kun Zhang.

**Formal analysis:** Kun Zhang, Gui-Quan Sun, Marco Ajelli, Keisuke Ejima, Quan-Hui Liu.

**Investigation:** Kun Zhang, Zhichu Xia, Shudong Huang, Gui-Quan Sun, Jiancheng Lv, Marco Ajelli, Keisuke Ejima, Quan-Hui Liu.

**Methodology:** Kun Zhang.

**Visualization:** Kun Zhang.

**Writing – original draft:** Kun Zhang, Keisuke Ejima, Quan-Hui Liu.

**Writing – review & editing:** Kun Zhang, Zhichu Xia, Shudong Huang, Gui-Quan Sun, Jiancheng Lv, Marco Ajelli, Keisuke Ejima, Quan-Hui Liu.

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
