## [Decision Letter · Decision Letter 0]

16 May 2023

Dear Liu,

Thank you very much for submitting your manuscript "Evaluating the impact of Test-Trace-Isolate for COVID-19 management and alternative strategies" for consideration at PLOS Computational Biology.

As with all papers reviewed by the journal, your manuscript was reviewed by members of the editorial board and by several independent reviewers. In light of the reviews (below this email), we would like to invite the resubmission of a significantly-revised version that takes into account the reviewers' comments.

We cannot make any decision about publication until we have seen the revised manuscript and your response to the reviewers' comments. Your revised manuscript is also likely to be sent to reviewers for further evaluation.

Sincerely,

David S. Khoury

Academic Editor

PLOS Computational Biology

Thomas Leitner

Section Editor

PLOS Computational Biology

Reviewer's Responses to Questions

**Comments to the Authors:**

Reviewer #1: The authors implement an agent-based model of SARS-CoV-2 disease spread where the contact patterns of agents are governed by two separate contact layers. The first represents close, regular contacts and the second represents casual, sporadic contacts. They incorporate the effects of test-trace-isolate strategies on contact rates within this modelling framework and investigate the consequences under different scenarios in terms of outcomes averted (total infections and deaths) and the cost of implementation (peak numbers of daily tests and simultaneously isolating individuals). The authors performed sensitivity analysis for several key parameters.

Overall, the study seems robust. However, I have a number of suggestions for improvement.

Major

Please include more details in the methods section. Unless the authors are limited by word counts, I suggest shifting some of the details from the methods supplement into the main text.

It is not clear in the methods section that contact survey data are used to inform the contact rates. This is the first and only mention (line 144): “More detailed description of the survey data and network generation process is available in Supplementary Methods.” There is no description of the survey data in the methods so how can it be more detailed in the SM. Further, please provide a more detailed description of the contact survey design and resulting data structure. It is not sufficient to point to the reference. It must be at least be summarised here.

The first time the model is described as a stochastic agent-based model is in the Discussion. This type of detail must be included in the methods section.

Your network includes household contacts. Does your model incorporate household structure? If not, worth reflecting on the implications of this in the Discussion.

Please include some plots of new infections (and deaths potentially too) over time. This is partly to confirm that the model behaviour is as expected, but also the authors state that the simulations are run for 1 year. Does the epidemic in all scenarios, including in the sensitivity analyses, “complete” within 1 year? It would also be useful to visualise for the reactive distancing scenarios how the epidemic trajectory changes with the addition of social restrictions.

In the discussion, please discuss your work in the context of other studies. You started to do this (very briefly) in the Results (line 197) but this should be done more comprehensively in the discussion. E.g. how do your findings, including the relationship between different parameters and outcomes, compare to other modelling studies e.g., https://doi.org/10.1016/S1473-3099(20)30361-3

You mention various limitations in the discussion. Please elaborate by speculating on how these limitations may have impacted your results.

Minor

I suggest stating how your study fits into the epidemiological and policy context of the COVID-19 pandemic. The baseline R0 of 2.5 and other features (fully susceptible population??) of your analysis suggest that the study is focused on the era of ancestral SARS-CoV-2 virus (I.e. before the emergence of the Alpha variant and availability of vaccines).

I suggest deleting the first line of the introduction as this statement is not true.

Line 52. I suggest deleting “and the etiology was not clear” as this statement is not true. The etiology was clear in March 2020 when many countries first implemented lockdowns.

Line 52. “…due to the huge impacts of the lockdowns on the functioning of the society” AND the availability of other control measures such as vaccines.

Line 76. What do you mean by “silent transmission”?

Line 79. Remove “etc” or explain what it is referring to.

Line 177. The first time SEIR is mentioned, it should be spelled out in full.

For visualising the contact rates between age groups, it would be useful to see plots of the contact matrices for both the static and temporal contact layers.

Line 145. The average numbers of contacts. Is this per day? 20+ seems quite high. Is this consistent with the survey data?

How did you initialise the simulations? Is the population fully susceptible?

Line 218. Typographical error. Captured by “contact tracing”.

Lines 218-219. Consider swapping “chance” with “probability”.

Line 237 - what do you mean by “better implemented”?

Line 356. I would strengthen this sentence. Delays are known to impact contact tracing effectiveness and have previously been investigated in modelling studies e.g. https://doi.org/10.1016/S2468-2667(20)30157-2

In the introduction, the authors mention that few studies of TTI have considered network structure. Please review this study by Firth et al “Using real-world network to model localised COVID-19 control strategies” . It seems relevant and should warrant a mention in the introduction and discussion. 10.1038/s41591-020-1036-8.

Worth noting in the discussion that you have used pre-pandemic contact data and it’s likely that contact rates and structures changed during the pandemic (even in the absence of government advice or mandated social restrictions). Indeed, this has been reported in many settings, and is evident through mobility data such as from Google. Further note, contact rates are most likely to spontaneously reduce when the perceived risk of infection is higher (i.e. as incidence increases). Therefore, the reactive distancing scenarios explored in your analysis need not only represent government policy scenarios, they could represent spontaneous changes in population behaviour.

Line 365. What do you mean by “highly infectious characteristics”? High transmissibility?

Reviewer #2: I want to start by thanking the authors for the submission. Like many of us, I have read a huge amount of COVID modelling papers but this one brings novelty and methodological advancement which has been an enjoyable read.

The paper is well written and I believe methodologically sound.

Introduction

I would like to see a little bit more background on using networks in infectious diseases modelling (for example Bansal 2010: https://www.tandfonline.com/doi/full/10.1080/17513758.2010.503376) just to highlight (what I think is) the novelty of the two layers and how the TTI system is implemented only in one layer.

It's quite unclear what the "cost" that is mentioned and calculated several times is. I particularly note the title for Figure 3 talks about a cost but the panels of Figure 3 are all epidemiological indicators (cases, deaths, number of contacts traced). My guess is these all could have a dollar value assigned to them, but if this is the case that needs to be included in the main manuscript.

Also on cost, are false positives included in the TTI scheme? From a dynamical systems perspective it doesn't matter (other than contacts being isolated when they don't need to be), but a false positive also means more contacts to be traced, and presumably more tests to be run, each of which costs money and will affect cost-benefit analysis.

Conclusion

I'd like to see a comment saying how effective TTI can be (or that it can't fully control highly infectious conditions) in here, at the moment it reads quite generically.

Minor comments:

Line 94: However, most of modelling studies -> most modelling studies

**Have the authors made all data and (if applicable) computational code underlying the findings in their manuscript fully available?**

Reviewer #1: **No: **Data and code were not available for review.

Reviewer #2: **No: **Authors state data and code will be made available upon acceptance, but it is not currently available.

PLOS authors have the option to publish the peer review history of their article (what does this mean?). If published, this will include your full peer review and any attached files.

Reviewer #1: No

Reviewer #2: **Yes: **Michael J Lydeamore
---

## [Decision Letter · Decision Letter 1]

9 Aug 2023

Dear Liu,

We are pleased to inform you that your manuscript 'Evaluating the impact of Test-Trace-Isolate for COVID-19 management and alternative strategies' has been provisionally accepted for publication in PLOS Computational Biology.

Best regards,

David S. Khoury

Academic Editor

PLOS Computational Biology

Thomas Leitner

Section Editor

PLOS Computational Biology

Reviewer's Responses to Questions

**Comments to the Authors:**

Reviewer #2: All requested changes have been made, I have nothing to add.

**Have the authors made all data and (if applicable) computational code underlying the findings in their manuscript fully available?**

Reviewer #2: None

PLOS authors have the option to publish the peer review history of their article (what does this mean?). If published, this will include your full peer review and any attached files.

Reviewer #2: **Yes: **Michael J Lydeamore

---

## [Editor Report · Acceptance letter]

25 Aug 2023

PCOMPBIOL-D-23-00428R1 

Evaluating the impact of Test-Trace-Isolate for COVID-19 management and alternative strategies

Dear Dr Liu,

I am pleased to inform you that your manuscript has been formally accepted for publication in PLOS Computational Biology. Your manuscript is now with our production department and you will be notified of the publication date in due course.

With kind regards,

Zsuzsanna Gémesi
